# Erlotinib Promotes Ligand-Induced EGFR Degradation in 3D but Not 2D Cultures of Pancreatic Ductal Adenocarcinoma Cells

**DOI:** 10.3390/cancers13184504

**Published:** 2021-09-07

**Authors:** Nausika Betriu, Anna Andreeva, Carlos E. Semino

**Affiliations:** Tissue Engineering Research Laboratory, Department of Bioengineering, IQS-School of Engineering, Ramon Llull University, 08017 Barcelona, Spain; nausikabetriur@iqs.url.edu (N.B.); annaandreeva@iqs.url.edu (A.A.)

**Keywords:** EGFR, trafficking, degradation, self-assembling peptides, 3D culture, pancreatic ductal adenocarcinoma, PDAC, drug resistance

## Abstract

**Simple Summary:**

The EGFR is a tyrosine kinase receptor that responds to different stresses such as UV irradiation, hypoxia and drug treatment by internalizing into endosomal compartments. Receptor trafficking and degradation due to tyrosine kinase inhibitors has been widely studied in two- dimensional (2D) cell culture systems, but little is known about how cells respond to these types of drugs in more physiologically relevant models such as three-dimensional (3D) cultures, whose nanostructured properties allow cells to grow, proliferate, migrate and extend cellular processes in their 3D space. In this study, we show that EGFR suffers degradation in response to erlotinib treatment in 3D-cultured cancer cells but not in classic 2D culture systems, demonstrating that dimensionality strongly affects cell drug response. This 3D model may pave the way for the development of more physiological culture platforms to obtain mechanistic insights into how cells respond to chemotherapy.

**Abstract:**

The epithelial growth factor receptor (EGFR) is a tyrosine kinase receptor that participates in many biological processes such as cell proliferation. In addition, EGFR is overexpressed in many epithelial cancers and therefore is a target for cancer therapy. Moreover, EGFR responds to lots of stimuli by internalizing into endosomes from where it can be recycled to the membrane or further sorted into lysosomes where it undergoes degradation. Two-dimensional cell cultures have been classically used to study EGFR trafficking mechanisms in cancer cells. However, it has been widely demonstrated that in 2D cultures cells are exposed to a non-physiological environment as compared to 3D cultures that provide the normal cellular conformation, matrix dimensionality and stiffness, as well as molecular gradients. Therefore, the microenvironment of solid tumors is better recreated in 3D culture models, and this is why they are becoming a more physiological alternative to study cancer physiology. Here, we develop a new model of EGFR internalization and degradation upon erlotinib treatment in pancreatic ductal adenocarcinoma (PDAC) cells cultured in a 3D self-assembling peptide scaffold. In this work, we show that treatment with the tyrosine kinase inhibitor erlotinib promotes EGFR degradation in 3D cultures of PDAC cell lines but not in 2D cultures. We also show that this receptor degradation does not occur in normal fibroblast cells, regardless of culture dimensionality. In conclusion, we demonstrate not only that erlotinib has a distinct effect on tumor and normal cells but also that pancreatic ductal adenocarcinoma cells respond differently to drug treatment when cultured in a 3D microenvironment. This study highlights the importance of culture systems that can more accurately mimic the in vivo tumor physiology.

## 1. Introduction

The epithelial growth factor receptor (EGFR) is a tyrosine kinase receptor (TKR) that participates in many biological processes such as cell proliferation, differentiation and motility, under both physiological and pathological conditions. Overexpression and/or hyperactivation of the EGFR is a hallmark of many epithelial cancers such as breast, lung, colon and pancreatic cancer, and therefore EGFR is a target for cancer treatment. EGFR inhibitors can be classified as small molecules tyrosine kinase inhibitors (such as erlotinib, gefitinib and lapatinib) or monoclonal antibodies (such as cetuximab and panitumumab) [1]. 

Autophosphorylation and activation of the EGFR is triggered by ligand binding, which initiates signaling cascades at the plasma membrane. Complete activation of the EGFR as well as termination of its signaling depends on its internalization into endosomes and intracellular trafficking [2]. EGFR can also be internalized due to different stresses such as UV irradiation [3], hypoxia [4,5] and oxidative stress [6]. The internalization mechanism of EGFR as well as whether the receptor is subsequently degraded in lysosomes or recycled to the membrane will depend on the type of stimulus. For example, EGF binding induces ubiquitin-dependent lysosomal degradation of the receptor and a partial recycling to the membrane [7], while TGFα ligand induces endocytosis and a rapid recycling [7,8]. Other stresses such as UV radiation, serum starvation or cisplatin treatment trigger internalization and arrest in nondegradative endosomes [2]. Tyrosine kinase inhibitors (TKIs) have also been described to promote EGFR trafficking in cancer cells. For example, gefitinib induces EGFR endocytosis and non-degradative endosomal arrest [9] as well as mitochondrial translocation [10] in glioblastoma cell lines. However, EGFR internalization due to TKIs exposure may be cell type-dependent, since other works report that gefitinib inhibits endocytosis in non-small cell lung cancer cell lines [11]. In contrast, the monoclonal antibody cetuximab has been shown to induce EGFR degradation [12], its sorting to the endoplasmic reticulum and nucleus [13], and mitochondrial translocation of the truncated form EGFRvIII [12].

Two-dimensional cell cultures have been used for many years to study not only EGFR trafficking mechanisms in normal and cancer cells but also as a model to study cell physiology and pathophysiology in general. However, it has been widely demonstrated that 2D cultures do not recreate the microenvironment in which normal cells exist, nor the milieu of solid tumors. In such 2D conditions, cells are forced to adhere to a flat and stiff substrate, which results in morphological changes that ultimately modify cellular function [14]. Moreover, molecular gradients are not reproduced and cells along the 2D surface are exposed to the same nutrient, oxygen and drug levels, while cells within a tumor are exposed to a large gradient of concentration as molecules diffuse from blood vessels [15]. It is also important to note that ECM composition and configuration are strongly modified in 2D cultures, and consequently cells do not receive the proper signals that a normal ECM configuration provides [14]. Three-dimensional (3D) cancer cell models allow one to better mimic the tumor microenvironment and manipulate each component in order to study its implications in tumor progression [16,17]. In this sense, extracellular matrix analogs, also called scaffolds, have become very popular among researchers [18]. Biomaterial scaffolds permit not only cell–cell but also ECM–cell interactions and provide the chemical, physical and mechanical cues needed for cells to form tissue structures in vitro [14].

The selection of the type of scaffold is a key point when planning experiments, and different factors such as the application of the 3D model, the tumor etiology and the concrete step of tumor progression to be recreated should be considered. For example, scaffolds that have a natural origin, such as collagen, are typically used to study cancer cell migration and invasion [19,20,21]. On the other hand, polymeric scaffolds such as poly(vinyl alcohol), poly(ethylene oxide terephthalate) (PEOT) and poly(butylene terephthalate) (PBT) have been used to investigate the influence of the scaffold architecture on pancreatic cancer cell growth and behavior, thus permitting one to create stage-specific pancreatic cancer models [22]. Moreover, when working with 3D cell culture, it is important to interpret the obtained results in the context of each experimental design. For example, culture of PDAC cells in collagen matrices promotes epithelial-to-mesenchymal transition (EMT), while culturing the same cells in basement membrane extract gels at a matched stiffness promotes mesenchymal-to-epithelial transition (MET) [23]. Furthermore, not only the composition but also the stiffness of the matrix is an important parameter to adjust when culturing cells in three dimensions. Increased matrix stiffness is a hallmark of many cancers such as breast [24], colorectal [25] and pancreatic [26] cancers. PDAC is characterized by a dense and fibrotic stroma due to the production of abundant amounts of ECM (mostly collagens) by stromal pancreatic cells [27]. In consequence, PDAC tissue can be several folds stiffer than its healthy counterpart [26,28], and different 3D models using synthetic [29] and natural scaffolds [19] have been developed in order to study the effect of matrix stiffness on PDAC cells.

In this report, we present a new 3D cell model of EGFR trafficking and degradation in pancreatic ductal adenocarcinoma (PDAC), based on the synthetic self-assembling peptide RAD16-I as a biomaterial for cell culture. In a neutral pH, this peptide self-assembles into a nanofiber network (around 10 nm diameter and 50–200 nm pore size) that allows for the embedding of cells in a 3D environment [18]. The main advantage of self-assembling peptide scaffolds (SAPS) over natural matrices is that they do not suffer degradation in vitro and therefore allow for the maintenance of the same mechanical conditions (matrix stiffness) during culture time. Moreover, it allows one to establish both soft and stiff 3D environments by simply changing its final concentration. RAD16-I is a non-instructive matrix from the point of view of receptor recognition/activation, and therefore this synthetic matrix holds the cells in an inert 3D configuration until they produce ECM proteins and decorate their own physiological environment. RAD16-I has been widely used as a cell culture platform for different tissue engineering applications such as bone [30], cartilage [31], cardiac [32] and hair [33] tissue engineering, and it has also been used to develop 3D models of ovarian [34], breast [35] and pancreatic [36] cancers.

In this work, we describe for the first time that treatment with TKI erlotinib, together with EGF, promotes EGFR degradation in 3D cultures of pancreatic ductal adenocarcinoma (PDAC) cell lines but not in 2D cultures. Moreover, we show that EGFR degradation due to erlotinib treatment does not occur in normal fibroblast cells. This new 3D cell model may introduce new perspectives in the study of EGFR degradation and its implications in cancer therapy, in an environment that more accurately reproduces the in vivo conditions found in a tumor. 

## 2. Materials and Methods

### 2.1. 2D Cell Culture

The human pancreatic ductal adenocarcinoma cell lines BxPC-3 (EP-CL-0042, Elabscience, Houston, TX, USA) and PANC-1 (CRL-1469, ATCC, Manassas, VA, USA), and primary human normal dermal fibroblasts (hNDF) (C-12302, Promocell, Heidelberg, Germany), were cultured at 10,000 cells/cm^2^ for no more than 15 passages in DMEM (DMEM-HXA, Capricorn, Ebsdorfergrund, Germany) or RPMI (RPMI-XA, Capricorn) in the case of BxPC-3, supplemented with 10% Fetal Bovine Serum (FBS) (S1810; Biowest, Nuaillé, France), L-glutamine (X055, Biowest) and Penicillin/Streptomycin (P/S) (L0022, Biowest). Cultures were maintained at 37 °C and 5% CO_2_ in a humidified atmosphere.

### 2.2. 3D Cell Culture in the Self-Assembling Peptide Scaffold RAD16-I

The protocol for cell encapsulation into self-assembling peptide scaffolds has been previously described in detail [37]. The peptide RAD16-I (commercially available at 1% in water, PuraMatrix^TM^, 354250, Corning, New York, NY, USA) was diluted to a final concentration of 0.3% (*v*/*v*) in 10% (*w*/*v*) sucrose (S0389, Merck, St Louis, MO, USA) or maintained at 1% (stock) and sonicated for 30 min. Meanwhile, cells were harvested by trypsinization and resuspended to 4·10^6^ cells/mL in 10% (*w*/*v*) sucrose, which is an isotonic and non-ionic medium that avoids peptide spontaneous assembly during the encapsulation process. The cell suspension was then mixed with an equal volume of 0.3% or 1% RAD16-I peptide solution to obtain a mixture of 2·10^6^ cells/mL and 0.15% (soft) or 0.5% (stiff) RAD16-I. Next, 40 µL of cell/peptide suspension (80,000 cells) was loaded into wells of a 48-well plate previously filled with 500 µL of culture medium, which induced the peptide spontaneous self-assembly. The plate was left in the flow cabinet for 20 min to let the peptide gel and then placed in the incubator for 1 h. Medium was changed twice to favor the leaching of sucrose. 3D cultures were maintained in DMEM or RPMI supplemented with 10% FBS, L-glutamine, P/S at 37 ºC and 5% CO_2_ in a humidified atmosphere, and medium was changed three times per week. Cells were cultured for 6 days to ensure adaptation to the 3D environment before being incubated with drugs and/or processed for protein extraction, MTT assay or immunofluorescence staining.

### 2.3. Drug Incubation

For 2D assays, cells were seeded at 10,000 cells/cm^2^ in 48-well plates and drugs were added on the following day. For 3D assays, cells were cultured for 6 days before adding the drug. In both 2D and 3D conditions, cells were incubated with 50 µM erlotinib and 10 ng/mL EGF for 16 h. To inhibit lysosome and proteasome degradation, cells were pre-treated for 8 h before erlotinib treatment with 600 nM bafilomycin A1 (SML1661, Merck) or 10 µM MG-132 (M7449, Merck), respectively.

### 2.4. MTT Assay for Cell Viability and Proliferation 

MTT [3-(4,5-dimethylthiazol-2-yl)-2,5-diphenyltetrazolium bromide] (M5655, Merck) was used to assess cell viability in 2D and 3D cultures. To determine erlotinib IC_50_, cells in 2D cultures were seeded at 10,000 cells/cm^2^ and the drug was added the following day. For 3D cultures, cells were cultured for 6 days before adding erlotinib. In both cases (2D and 3D cultures), erlotinib IC_50_ was calculated at 72 h from drug addition using MTT assay. For that, cell culture medium was aspirated and 200 μL (for 2D cultures) or 500 μL (for 3D cultures) of MTT reagent were added to a final concentration of 0.5 mg/mL in culture medium. Samples were incubated for 2 h (2D cultures) or 3 h (3D cultures) at 37 ºC and 5% CO_2_ in a humidified atmosphere. MTT solution was then removed and cells were lysed with 200 µL of DMSO (D8418, Merck). Absorbance was read at 570 nm using a microplate reader (BiotekEpoch^TM^, Biotek, Winooski, VT, USA).

### 2.5. Immunofluorescence

Cells in 2D and 3D cultures were fixed with 3.7% formaldehyde for 15 min and washed with 1x PBS. Cultures were blocked with 5% BSA/0.1% Triton X-100 in PBS for 1 h (for 2D cultures) or 2 h (3D cultures) and incubated overnight at 4ºC with the following primary antibodies: anti-EGFR (700308, Invitrogen, Waltham, MA, USA) at 1:100, anti-integrin β1 (ab24693, abcam, Cambridge, UK) at 1:500, anti-LAMP1 (14-1079-80; Invitrogen) at 1:150 and anti-EEA1 (14-9114-80, Invitrogen) at 1:500 dilution in 1% BSA. Next, cells were washed with 1% BSA and incubated for 2 h with secondary antibodies conjugated with Alexa Fluor 488 (ab150105, abcam) and 647 (ab150079, abcam) at 1:500. Finally, cells were counterstained with Phalloidin-TRITC and DAPI for cytoskeleton and nuclei visualization. Pictures were acquired with Leica Thunder Imager widefield microscope (Leica Microsystems, Wetzlar, Germany) coupled to a Leica DFC9000 GTC sCMOS camera, using an APO 63x objective. Images were processed and analyzed with ImageJ software version 2017-05-30 (NIH, Bethesda, MD, USA) [38]. 

### 2.6. Image Analysis 

For colocalization analysis, Manders’ coefficients were calculated. Manders’ coefficients are an overlapping parameter that describe the proportion of channel A signal coinciding with channel B over the total A intensity (M_1_) and the proportion of channel B signal coinciding with channel A over the total B intensity (M_2_) [39]. This coefficient ranges from 0 (no overlapping) to 1 (total overlapping). Manders’ coefficients are very sensitive to noise, and for this reason, background needs to be set to zero. Moreover, to calculate Mander’s coefficients it is important to establish a threshold for segmentation. The analysis was performed using ImageJ software [38]. Each channel was processed for background subtraction and filtered using Median and Gaussian Blur to reduce the presence of noise. Images were then segmented by thresholding using the Default option, and the resulting binary images were cleaned with the Erode and Open functions. Binary images were then used as masks to sample the denoised images using the Image Calculator function with the Min operator, creating a background-less image for each channel [40]. Finally, Manders’ colocalization coefficients were calculated using JACoP (Just Another Colocalization Plugin) version 2.0 [41] by setting the threshold values to 1. Coefficients were obtained from at least 5 images (*n*>5) containing 5–10 cells per colony in the case of tumor cells or single cells in hNDF.

### 2.7. Western Blot

2D cultures and 3D constructs were lysed with RIPA buffer (R0278, Merck) containing protease inhibitor cocktail (11836153001, Roche, Basel, Switzerland). Total protein content was quantified with a BCA protein assay kit (39228, Serva, Heidelberg, Germany) and 5 or 10 μg of protein were loaded into 8% polyacrylamide gels and run by applying 225 V for 40 min. Afterwards, proteins were transferred to a PVDF membrane (IPVH07850, Merck) by applying 40 V for 2 h. The membrane was then blocked for 1 h with 4% (*w*/*v*) nonfat powdered milk in 0.2% PBS-Tween. Next, the membrane was incubated with primary antibodies anti-EGFR (700308, Invitrogen) at 1:1000 and anti-GAPDH (649201, Biolegend, San Diego, CA, USA) at 1:2000 for 1 h at room temperature. The membrane was then washed and incubated with secondary antibodies anti rabbit-HRP (ab6721, abcam) and anti mouse-HRP (ab6820, abcam), both at 1:1000 for 1 h at RT. Finally, the membrane was revealed for HRP detection with a SuperSignal West Pico Chemiluminescent Substrate (34080, Thermo Fisher Scientific, Waltham, MA, USA). Chemiluminescent images were taken in the ImageQuant^TM^ LAS 4000 mini (GE HealthCare, Chicago, IL, USA). Protein bands were quantified using ImageJ software and expressed as a ratio between the protein of interest and the loading control. Each blot was repeated three times (N = 3).

### 2.8. Statistics

Data are presented as mean ± Standard Deviation. Conditions were tested in triplicate (*n* = 3) in three independent experiments (N =3). Statistical differences were analyzed with GraphPad Prism 6 (San Diego, CA, USA) by one-way or two-way ANOVA followed by Tukey’s multiple comparisons test. Statistical differences were indicated as * for *p* < 0.05, ** for *p* < 0.01, *** for *p* < 0.001 and **** for *p* < 0.0001.

## 3. Results

### 3.1. Cell Culture in RAD16-I Scaffold

In the present work, the pancreatic cancer cell lines BxPC-3 and PANC-1 were used, as well as primary human dermal fibroblasts (hNDF), as a control of healthy, non-tumor cells, to study the effect of the tyrosine kinase inhibitor (TKI) erlotinib on EGFR internalization and degradation in a three-dimensional (3D) environment. These two PDAC cell lines were chosen for three main reasons. First, they represent a model of epithelial phenotype (BxPC-3) and an intermediate epithelial-mesenchymal phenotype (PANC-1) [23]. Second, they present with differences regarding erlotinib sensitivity, with BxPC-3 being considered an erlotinib-sensitive line and PANC-1 an insensitive one [42]. Third, they present with different KRAS genetic signatures, an important gene involved in PDAC progression. BxPC-3 cells have wild type KRAS, while PANC-1 cells present with a G12C mutation that produces its constitutive activation. This is of vital importance since 95% of primary pancreatic tumors show mutations in the KRAS gene [43]. For three-dimensional cell culture, we used the self-assembling peptide scaffold RAD16-I, which has been previously used for different cancer cell line cultures [34,35,36]. Moreover, cultures in RAD16-I were prepared at two different peptide concentrations: 0.15% and 0.5%, which correspond to a stiffness of 120 Pa (namely, soft matrix) and 1500 Pa (namely, stiff matrix), respectively, previously measured by rheometry [44]. Results show that both pancreatic cancer cell lines presented good viability regardless of matrix stiffness (Figure 1a). Moreover, growth rate in 3D cultures decreased for both cell lines compared to 2D, partially recapitulating the growth behavior of in vivo cancer cells (Table 1). Dermal fibroblasts did not show proliferation in the soft RAD16-I matrix, as previously reported [36], but did proliferate in the stiff matrix (Table 1). This behavior was previously shown for fibroblasts cultured in 3D collagen gels [45]. We also determined cell viability after 72 h incubation with the TKI erlotinib. Under our culture conditions, we determined an IC_50_ of 10 µM for the erlotinib-sensitive cell line BxPC-3 and for normal fibroblasts, in both 2D and 3D cultures. For the erlotinib-insensitive cell line PANC-1, we determined an IC_50_ of 45 µM for 2D cultures and 100 µM for 3D cultures, in both matrix stiffness conditions (Table 1, Appendix A). Therefore, the 3D environment promoted these cells to be even more resistant to erlotinib treatment.

Regarding cell phenotype, the epithelial cell line BxPC-3 grew forming round or oval clusters (Figure 1b, left), while PANC-1 cells formed more grape-like spheres (Figure 1b, middle). We did not find a correlation between matrix stiffness and cell phenotype or colony size in the tumor cells analyzed. Matrix stiffness had a great effect on dermal fibroblasts, which formed a highly interconnected network in the soft matrix, contracting the hydrogel and notably reducing its dimension in a few days (Figure 1c). In the stiff matrix, fibroblasts managed to interconnect to each other as well (Figure 1b, right) but required more time in culture to do so and contracted the matrix to a lesser extent (Figure 1c). This hydrogel-contraction behavior (also called matrix condensation) is characteristic of primary mesenchymal cells cultured in this kind of scaffold [33] but did not happen in tumor cells (Figure 1d).

### 3.2. EGFR Expression in 2D and RAD16-I 3D Cultures

We next analyzed the location and expression levels of the EGFR by immunofluorescence and western blot in both culture types. In 2D cultures of BxPC-3 cells, the EGFR displayed a strong staining in the cell periphery, colocalizing with β1-integrin, as well as a diffused cytoplasmatic staining (Figure 2a, top). In PANC-1 cells, the EGFR was found mainly in the cell periphery, presenting a strong staining that overlapped with β1-integrin (Figure 2a, middle). In fibroblasts instead, the EGFR showed a dotted staining all over the cell (Figure 2a, bottom). Similar to 2D cultures, BxPC-3 and PANC-1 in 3D cultures showed a peripheral staining of the EGFR, while hNDF displayed a dotted staining (Figure 2b). Western blot analysis revealed that total EGFR was downregulated in 3D cultures compared to 2D monolayer cultures in the three types of cells analyzed (Figure 2c), especially in dermal fibroblasts. Moreover, matrix stiffness in 3D cultures also influenced EGFR expression, being downregulated in stiff conditions compared to soft cultures in hNDF cells but not in BxPC-3 and PANC-1, which presented similar EGFR levels regardless of the stiffness (Figure 2d). 

### 3.3. Effect of EGF and Erlotinib on the Location of the EGFR

We next analyzed the effect of EGF and acute erlotinib treatment on cells in 2D and 3D cultures. For that, cells were incubated with 10 ng/mL EGF or 50 µM erlotinib or a combination of both during 16 h. Under control conditions, EGFR in 2D-cultured PANC-1 cells was found predominantly in the cell membrane, showing a strong peripheral staining (Figure 3a,b) and colocalizing with β1-integrin (Figure 3c). The presence of EGF triggered ligand-induced endocytosis of the EGFR, as previously described [7], thus being internalized from the cell periphery (Figure 3a, white arrows) into the cytoplasm (Figure 3a, empty arrows), accumulating perinuclearly (Figure 3a,b). Treatment of cells with erlotinib did not induce EGFR internalization, as demonstrated by its peripheral staining (Figure 3a,b) and high colocalization degree with β1-integrin, similar to control conditions (Figure 3c). However, when we incubated the cells with erlotinib in combination with EGF, we detected both membrane and perinuclear EGFR staining (Figure 3a,b).

The location of EGFR in 2D cultures of BxPC-3 under control conditions was mainly peripheral and was associated with β1-integrin staining, and diffused expression was detected in the cytoplasm (Figure 4a). The presence of EGF induced a strong accumulation of the receptor in the cytoplasm and the perinuclear area (Figure 4a,b). In the presence of erlotinib, EGFR displayed a similar location as in the control (Figure 4a), but the fraction of β1-integrin colocalizing with EGFR increased compared to the control (Figure 4c). Finally, when BxPC-3 cells were incubated with erlotinib in combination with EGF, we detected strong perinuclear staining (Figure 4a,b) but also an increased colocalization degree with the membrane marker β1-integrin compared to EGF-incubated cells (Figure 4c), similar to what happened in PANC-1 cells. The cytoplasmatic staining of EGFR in BxPC-3 under control conditions suggests that basal levels of EGFR trafficking may exist, which could be for different reasons. First, even though EGFR mainly resides in the plasmatic membrane, it constantly undergoes trafficking through the endocytic system [2]. Second, it is important to note that in order to keep culture conditions between 2D and 3D as similar as possible, cells were not serum-starved prior to the experiments. In 2D cultures, FBS proteins are removed by simply changing the culture medium. However, protein release from RAD16-I hydrogels can take more than 50 h [46,47], and therefore complete serum depletion is not possible in the short term in our 3D culture system. 

Finally, we analyzed the effect of EGF and erlotinib in 2D cultures of human normal dermal fibroblasts (hNDF). In this case, cells under control conditions showed a punctate staining of the EGFR distributed all over the cell and also some pericellular staining (Figure 5). The presence of EGF alone or combined with erlotinib induced the accumulation of the receptor mainly in the perinuclear area, probably in endosomal compartments, while erlotinib alone promoted the accumulation of the receptor along all the cell surface and pericellularly (Figure 5).

Altogether, these results suggest that erlotinib partially prevents EGFR internalization and trafficking to endosomal compartments upon EGF binding in 2D cultures, retaining part of the receptor in the plasmatic membrane, as previously described [48].

In 3D cultures, both tumor cell lines displayed peripheral staining of the EGFR, as happened in 2D cultures (Figure 6, top and middle). Under control conditions and in the presence of erlotinib alone, EGFR staining was peripheral and well-defined, but when EGF or EGF and erlotinib were added, EGFR signal in the membrane became more diffused and both pericellular and cytoplasmatic punctate staining were detected, which showed evidence of the internalization of the receptor (Figure 6, see arrows). In hNDF under control conditions, the EGFR displayed a punctate staining all over the cell while accumulating perinuclearly after EGF exposure (Figure 6, bottom, see arrows). In erlotinib-treated hNDF, the dotted EGFR expression found in the control was lost, becoming strongly concentrated all over the cell. When combining EGF and erlotinib treatment, both phenotypes could be detected.

### 3.4. Effect of EGF and Erlotinib on EGFR Degradation

It has been previously shown using 2D cultures that treatment of cells with monoclonal antibodies targeting the EGFR such as cetuximab [12] and Sym004 [48,49] resulted in overall decrease of EGFR levels due to protein degradation. In contrast, TKI treatment has not been shown to induce significant EGFR degradation in different 2D-cultured cancer cell lines [50,51,52,53]. Consistent with these studies we did not detect a relevant decrease in EGFR levels after 16 h erlotinib treatment (neither alone nor in combination with EGF) in any of the cells analyzed (Figure 7a,c, lanes 3–4, Appendix A). On the other hand, EGF treatment alone induced strong EGFR degradation in hNDF but not in tumor cells (Figure 7a,c, lane 2), which express much higher EGFR levels than hNDF. Therefore, it is likely that the EGF dose used (10 ng/mL) is insufficient to promote EGFR degradation in these PDAC cell lines. Remarkably, the presence of erlotinib in hNDF treated with EGF rescued EGFR levels similar to those in untreated cells (Figure 7a,c, lane 4), reinforcing the hypothesis that erlotinib prevents EGFR internalization in 2D-cultered cells.

Similar to 2D cultures, EGF did not induce significant EGFR degradation in 3D-cultured tumor cells. On the contrary, EGF induced a dramatic EGFR degradation in 3D cultures of fibroblasts regardless of matrix stiffness (Figure 7b,d, lanes 1–4, Appendix A). Interestingly, and contrary to what was found in 2D cultures, erlotinib promoted EGFR degradation when combined with EGF, but not alone, in PANC-1 and BxPC-3 cells cultured in 3D scaffolds under both stiffness conditions (Figure 7b,d, lanes 5–8). Moreover, the extent of EGFR degradation was erlotinib dose-dependent (Appendix A).

Contrary to tumor cells, the presence of erlotinib in EGF-incubated fibroblasts in 3D not only induced the degradation of the receptor but prevented it (Figure 7b,d, lanes 5–8), as happened in 2D cultures. In conclusion, erlotinib treatment (combined with EGF) had a contrary effect depending on cell type and dimensionality, promoting EGFR degradation in PDAC cell lines cultured in 3D but preventing it in normal fibroblasts in both 2D and 3D cultures.

To confirm that the decrease of EGFR levels in 3D cultures of PANC-1 and BxPC-3 cells treated with EGF and erlotinib was actually due to protein degradation, we inhibited proteasomal and lysosomal degradation by pre-incubating the cells with MG-132 (proteasome inhibitor) and bafilomycin A1 (lysosome inhibitor). It has been extensively reported that EGF-induced degradation occurs via lysosomes [7], and therefore we questioned whether EGFR downregulation in erlotinib-treated cells (Figure 7b,d) was also due to lysosomal degradation. 

Results show that lysosome inhibition in cells treated with erlotinib and EGF led to a notable increase in EGFR levels due to protein accumulation (Figure 8a,b, lane 8, Appendix A). Bafilomycin A1 inhibits fusion between endosomes and lysosomes, which causes the accumulation of cargo unable to suffer degradation [54,55]. Moreover, MG-132 also led to EGFR accumulation upon EGF and erlotinib treatment, but to a lesser extent than bafilomycin A1 (Figure 8a,b, lane 7). Proteasomal inhibition has been reported to deplete the free ubiquitin pool within the cell, thus interfering with protein degradation [56]. Moreover, combinatorial treatment with both inhibitors (MG-132 and bafilomycin A1) also led to protein accumulation in PANC-1 but not in BxPC-3 cells, which presented with reduced EGFR levels compared to its respective control (Figure 8a,b, lanes 5 and 9). This could be explained by the fact that proteasome disruption leads to endoplasmic reticulum stress to which the cell responds by attenuating protein translation, thereby inhibiting global protein synthesis [57]. Altogether, these results confirm that EGFR degradation due to erlotinib and EGF treatment in our 3D cancer cell culture system occurs via lysosomes.

### 3.5. EGFR Trafficking to Early Endosomes and Lysosomes in 2D and 3D Cultures

It is well established that after stimulation with EGF, the EGFR is activated, internalized and sorted into endosomal compartments. Once in EEA1-positive early endosomes, a small fraction of the receptor is recycled to the plasmatic membrane, while most of it is sorted into late endosomes and subsequently degraded in lysosomes [7]. In 2D cultures, we found a low EGFR fraction colocalizing with EEA1-positive early endosomes in PANC-1 (Figure 9a,d) and BxPC-3 tumor cells (Figure 9b,d). After EGF exposure, the fraction of EGFR colocalizing with EEA1 increased for both tumor cell lines (Figure 9d). Moreover, and in concordance with the hypothesis that erlotinib partially prevents EGFR internalization (Figure 3, Figure 4 and Figure 5), erlotinib-treated cells presented with the lowest colocalization values between EGFR and EEA1. Additionally, we found that in hNDF (Figure 9c) the EGFR fraction colocalizing with EEA1 was extremely low for all the conditions tested (Figure 9d), but on the contrary, Manders’ coefficients obtained for the proportion of EEA1-positive early endosomes containing EGFR signal were almost 1 (Figure 9e), meaning that almost all EEA1-positive endosomes were carrying EGFR (see Section 2.6). These differences between both Manders’ coefficients in hNDF exist because these are very large cells expressing EGFR all over the cellular milieu, while early endosomes are located mainly perinuclearly. Therefore, the EGFR signal coinciding with EEA1 signal over the total EGFR intensity (EGFR vs. EEA1) is much lower than the EEA1 signal coinciding with EGFR signal over the total EEA1 intensity (EEA1 vs. EGFR). Moreover, even colocalization between EEA1 and EGFR (Figure 9e) was similar between the control and the EGF-incubated cells, it significantly decreased in the presence of erlotinib alone, indicating a basal EGFR trafficking that was prevented by the presence of erlotinib. 

We next analyzed colocalization between EGFR and LAMP1 (lysosomal-associated membrane protein 1), a well-known lysosomal marker. Lysosome distribution was mainly perinuclear in BxPC-1 and hNDF cells under all conditions tested (Figure 10b,c). However, in PANC-1 cells, lysosomes were distributed all over the cytoplasm in control conditions as well as when incubated with erlotinib but accumulated perinuclearly after EGF treatment (Figure 10a). LAMP1 staining showed significantly higher colocalization coefficients between lysosomes and the EGFR in cells treated with EGF compared to control in all the cell types analyzed (Figure 10d,e). Moreover, the presence of erlotinib alone produced significant differences only in BxPC-3 cells, in which Manders’ coefficients (LAMP1 vs. EGFR) (Figure 10e) increased compared to the control.

Finally, we analyzed EGFR sorting into early endosomes and lysosomes in 3D cultures. In BxPC-3 tumor cells, we detected EGFR trafficking to EEA1-positive early endosomes in all conditions tested, but colocalization was much more evident in cells treated with EGF or EGF and erlotinib (Figure 11a, see arrows), in which EGFR was undergoing endocytosis. A similar pattern was found in hNDF (Figure 11b, see arrows). Regarding EGFR sorting into lysosomes, we found that in PANC-1 (Figure 12a) and BxPC-3 cells (Figure 12b), colocalization between EGFR and LAMP1 was mostly detected in cells treated with EGF or EGF and erlotinib. Even though we detected certain colocalization between EGFR and LAMP1-positive lysosomes in 3D-cultured tumor cells incubated with EGF, degradation under these conditions was undetectable at a western blot level (Figure 7), suggesting that the presence of erlotinib exacerbated this degradation in EGF-treated cells.

## 4. Discussion

The EGFR is the unique TKR reported to respond to multiple stimuli by internalizing into endosomal compartments. Moreover, different stressors induce different trafficking pathways that can end in endosomal accumulation and recycling to the membrane or degradation. Two-dimensional cell cultures are generally used to study EGFR trafficking mechanisms because they are economically affordable and easier to handle as well as to analyze than 3D cultures. However, it is widely accepted that 2D cultures do not recreate the microenvironment of in vivo tissue cells nor can they predict therapy outcome as precisely as 3D cultures do. Growing cells in a 3D environment reveals a more realistic drug response, being that 3D-cultured cells are more resistant to chemotherapy when compared to the same cells grown in 2D monolayer [34,58,59]. Different mechanisms have been attributed to this enhanced drug resistance, but the most straightforward explanation is that the microenvironment provided by the 3D system protects the cells from drug penetration [60]. Besides, most drugs have rapidly dividing cells as a target, and 3D cultures have been described to decrease the proliferation rate of cancer cells compared to 2D cultures [36]. Moreover, quiescent cells that exist in the inner part of the 3D culture remain protected from drug effect [61]. In addition, some pancreatic cancer cell lines have shown an increased expression of diverse drug resistance genes in 3D culture compared to 2D culture [62]. For example, PDAC cells in 3D collagen hydrogels but not in 2D cultures present with gemcitabine resistance through the upregulation of the membrane type-I matrix metalloprotease (MT1-MMP) [63]. Therefore, 3D-cultured cells can recapitulate mechanisms of drug resistance found in tumors, thus offering the opportunity to analyze these mechanisms and test multidrug therapies in vitro in order to reduce animal experimentation.

In this paper, we report a new model for EGFR internalization and degradation due to tyrosine-kinase inhibitor treatment in a 3D cell culture of pancreatic ductal adenocarcinoma cells. Using the synthetic self-assembling peptide RAD16-I as a platform for 3D culture, we found that erlotinib treatment combined with EGF but not alone promoted EGFR degradation in 3D but not 2D cultures of both PDAC cell lines (Figure 7). These results suggest that in 3D culture, EGF is necessary to promote EGFR endocytosis in PDAC cell lines, and that upon internalization, erlotinib promotes its lysosomal degradation (Figure 8). Interestingly, this behavior was not detected in normal fibroblasts, in which erlotinib prevented EGF-induced EGFR internalization and degradation in both 2D and 3D cultures (Figure 5, Figure 6 and Figure 7). In this sense, results can be controversial since previous in vitro studies demonstrate that TKI inhibitors such as erlotinib and gefitinib induce EGFR internalization and accumulation in non-degradative endosomes in different cell types [9], while others describe that the same TKI suppresses ligand-stimulated endocytosis [11]. Another work in which the authors develop an in vivo tumor model of human oral squamous carcinoma using xenografts reports that gefitinib treatment inhibits EGFR endocytosis [64], suggesting that receptor kinase activity is required for receptor internalization [65]. We indeed detected this kinase activity-dependence to internalize the receptor in fibroblasts (Figure 5) and in tumor cells, but to a lesser extent (Figure 3 and Figure 4). One explanation could be that PDAC cell lines are more resistant to erlotinib and therefore the dose used could be not enough to inhibit the kinase activity of the receptor. However, we used an acute dose of 50 µM, corresponding to the IC_50_ of the erlotinib-resistant cell line PANC-1; this should be enough to inhibit the EGFR in the erlotinib-sensitive BxPC-3 cell line and in hNDF, which both presented an IC_50_ of 10 µM. 

It is well established that receptor recycling to the plasmatic membrane after endocytosis leads to continuous signaling. By contrast, degradation in lysosomes is associated with signaling attenuation, and therefore endocytic downregulation could be associated with TKIs sensitization. For example, it has been reported that mutant-EGFR forms are internalized via clathrin-mediated endocytosis and sorted into endosomal compartments, where they continue signaling [66]. Instead, clathrin inhibition in mutant-EGFR-expressing cells induces micropinocytosis-dependent EGFR internalization followed by degradation in lysosomes, which results in signal extinction and apoptosis, and thus overcoming resistance to TKI [66]. Another recent study reports that TKI treatment induces intracellular accumulation of mutated-EGFR. The authors found a positive correlation between EGFR accumulation and clinical benefit in patients presenting with tumors harboring these EGFR mutations, suggesting that blocked EGFR-membrane recycling contributes to TKI sensitivity and a positive therapy outcome [67]. Our results demonstrate not only that erlotinib has a different effect on tumor and normal cells but also that PDAC cells respond differently to erlotinib treatment when cultured in a 3D microenvironment. In fact, PANC-1 cells increased their erlotinib resistance from 45 to 100 µM when cultured in 2D vs. 3D (Table 1), even though under these 3D conditions the EGFR was undergoing degradation. On the other hand, BxPC-3 cells did not show increased erlotinib resistance when cultured in 3D but also showed EGFR degradation. Therefore, under our experimental conditions, EGFR degradation did not sensitize the cells to erlotinib treatment. Given that this EGFR degradation was only detected in 3D cultures, it is possible that in this 3D environment PDAC cells acquire additional resistance mechanisms that allow them to survive independently of the EGFR signaling pathway. 

Importantly, EGFR degradation in 3D cultures due to TKI was only detected in the presence of EGF, suggesting that ligand binding is necessary for endocytosis and that once the receptor is internalized, erlotinib treatment promotes its degradation. EGF can reach very high concentrations in different body fluids such as bile, urine and milk [68,69], and a normal epithelium avoids these fluids to reach EGF receptors, which are expressed basolaterally. However, as a result of a premalignant neoplasia, in which the tight junctions of the epithelium become leaky, high concentrations of EGF can reach and activate EGFRs [70]. Moreover, since EGFR ligands have been found in several tumors [71,72,73] it is likely that our 3D model could closely mimic the tumor in vivo scenario during erlotinib therapy.

## 5. Conclusions

We have developed a 3D cancer cell culture system to study EGFR trafficking and degradation due to TKI treatment. This study highlights not only that erlotinib has a distinct effect on tumor and normal cells but also that pancreatic ductal adenocarcinoma cells respond differently to erlotinib treatment when cultured in a 3D microenvironment. To our knowledge, this is the first time where EGFR trafficking is explored in a three-dimensional environment, and it is our hope that this report may encourage other researchers to introduce 3D cell culture systems in the study of receptor trafficking. Moreover, future research might be focused in studying the effect that ECM components present in PDAC, such as collagens and hyaluronic acid, could have on EGFR trafficking.

## Figures and Tables

**Figure 1 cancers-13-04504-f001:**
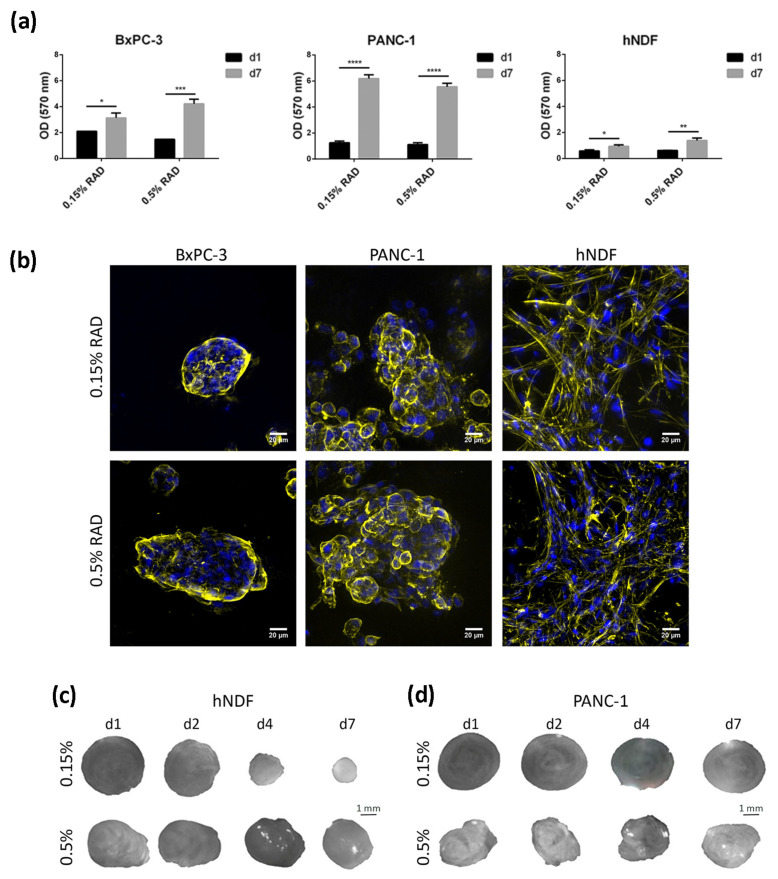
Cell culture in RAD16-I scaffold. (**a**)Viability of BxPC-3, PANC-1 and hNDF cultured in 0.15% and 0.5% RAD16-I scaffold measured by MTT assay at day 1 and day 7 of culture. Statistical differences are indicated as * for *p* < 0.05, ** for *p* < 0.01, *** for *p* <0.001 and **** for *p* < 0.0001, two-way ANOVA, N = 2, *n* = 3); (**b**) Z-projection pictures of BxPC-3, PANC-1 and hNDF cells at day 6 of culture stained with Phalloidin (pseudo-colored in yellow) and DAPI (blue). Scale bars represent 20 µm; (**c**) macroscopic view of hNDF and (**d**) PANC-1 3D constructs in 0.15% and 0.5% RAD16-I scaffolds at different time points.

**Figure 2 cancers-13-04504-f002:**
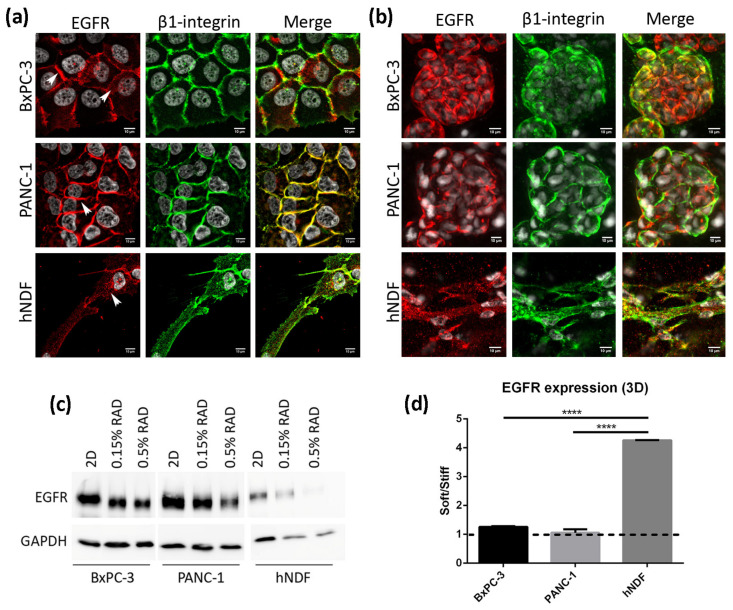
EGFR expression in 2D and 3D cultures. (**a**) EGFR (red) and β1-integrin (green) immunofluorescence counterstained with DAPI (grey) in BxPC-3, PANC-1 and hNDF cells cultured in 2D monolayer and (**b**) Z-projection of EGFR and β1-integrin immunofluorescence in BxPC-3, PANC-1 and hNDF cells in 3D RAD16-I scaffold at 0.15% peptide concentration. Scale bars represent 10 µm; (**c**) Western blot bands of EGFR in 2D and 3D cultures; (**d**) quantification of EGFR in 3D cultures represented as the ratio between soft and stiff cultures. GAPDH was used as loading control. One representative blot is shown. Experiments were repeated three times (N = 3), and statistical differences are indicated as **** for *p* < 0.0001.

**Figure 3 cancers-13-04504-f003:**
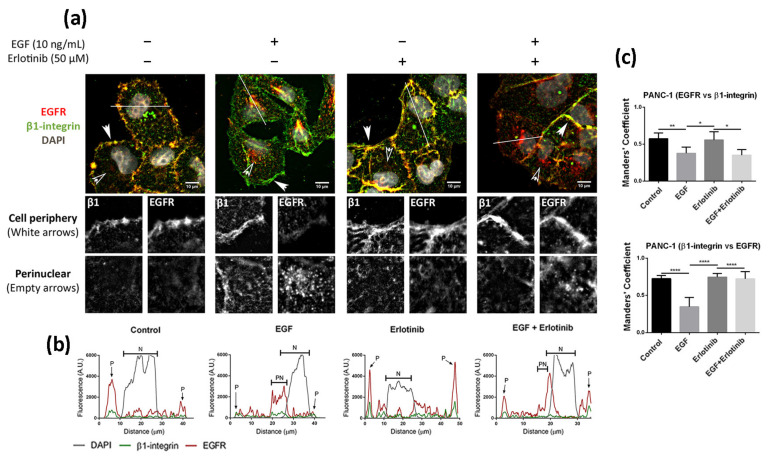
Immunofluorescence analysis of the EGFR in PANC-1 cells incubated with EGF, erlotinib or both in 2D cultures. (**a**) EGFR (red) and β1-integrin (green) immunofluorescence counterstained with DAPI (grey) in the presence of EGF, erlotinib or both in PANC-1 and close-up sections (gray pictures) of the cell periphery and the perinuclear area labeled with white and empty arrows, respectively. Scale bars represent 10 µm; (**b**) fluorescence intensity profiles corresponding to the white line in pictures from (**a**). Different cell regions are indicated as P for cell periphery, N for nucleus and PN for perinuclear area; (**c**) Manders’ colocalization coefficients. Statistical differences are indicated as * for *p* < 0.05, ** for *p* < 0.01 and **** for *p* < 0.0001.

**Figure 4 cancers-13-04504-f004:**
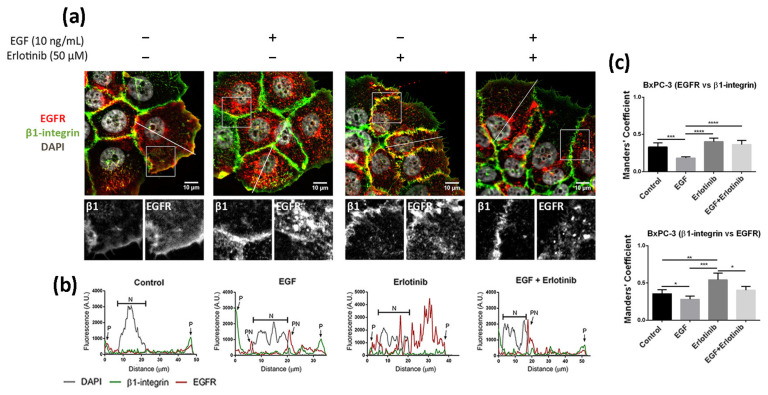
Immunofluorescence analysis of the EGFR in BxPC-3 cells incubated with EGF, erlotinib or both in 2D cultures. (**a**) EGFR (red) and β1-integrin (green) immunofluorescence counterstained with DAPI (grey) in the presence of EGF, erlotinib or both and close-up sections (gray pictures). Scale bars represent 10 µm; (**b**) fluorescence intensity profiles corresponding to the white line in pictures from (**a**). Different cell regions are indicated as: P for cell periphery, N for nucleus and PN for perinuclear area; (**c**) Manders’ colocalization coefficients. Statistical differences are indicated as * for *p* < 0.05, ** for *p* < 0.01, *** for *p* < 0.001 and **** for *p* < 0.0001.

**Figure 5 cancers-13-04504-f005:**
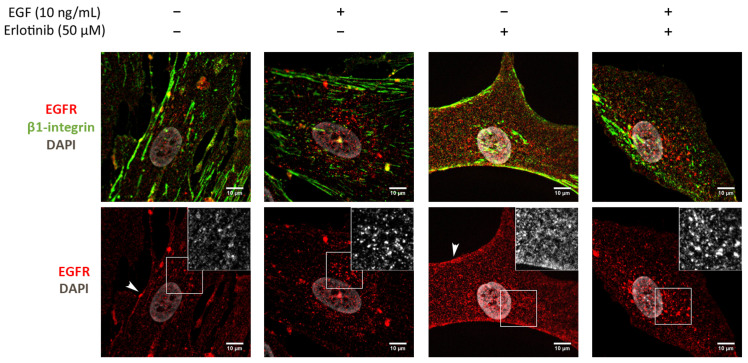
Immunofluorescence analysis of the EGFR in hNDF cells incubated with EGF, erlotinib or both in 2D cultures. Scale bars represent 10 µm. Insets represent high magnification images of the region indicated by a white square.

**Figure 6 cancers-13-04504-f006:**
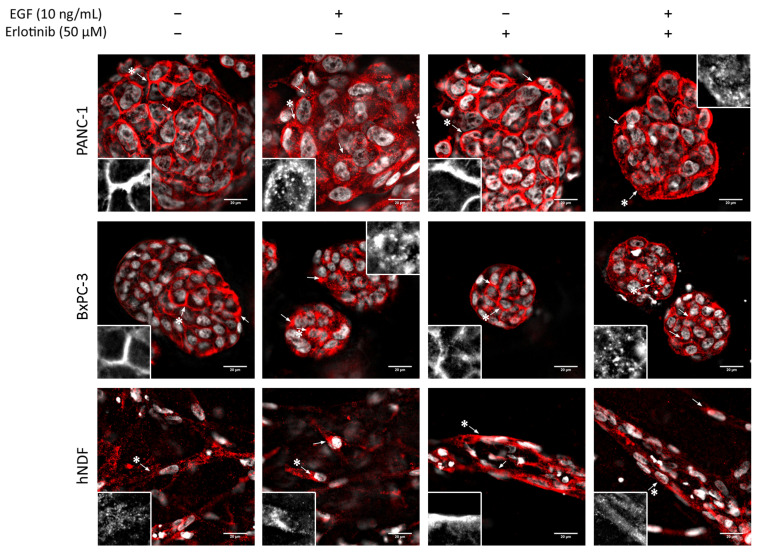
EGFR immunofluorescence in PANC-1, BxPC-3and hNDF cells incubated with EGF, erlotinib or both in 0.15% RAD16-I 3D cultures. One representative Z plane is shown. Scale bars represent 20 µm. Insets represent high magnification images of the region indicated by an asterisk (*).

**Figure 7 cancers-13-04504-f007:**
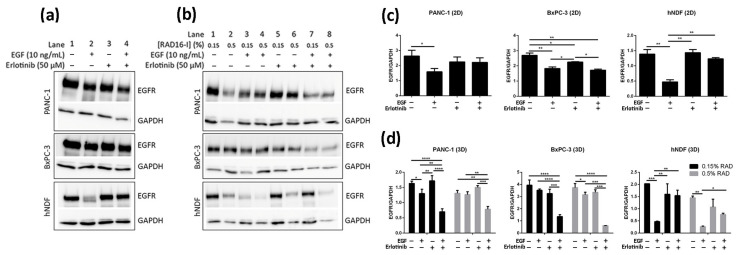
Western blot analysis of the EGFR in BxPC-3, PANC-1 and hNDF cells incubated with EGF or erlotinib or both in 2D cultures and soft and stiff 3D cultures. (**a**) Western blot bands of EGFR in 2D cultures; (**b**) Western blot bands of EGFR in 0.15% and 0.5% RAD16-I 3D cultures; (**c**) densitometry of bands shown in (**a**) for 2D cultures; (**d**) densitometry of bands shown in (**b**) for 3D cultures. GAPDH was used as an internal control. One representative blot is shown. Experiments were repeated three times (N = 3), and statistical differences are indicated as * for *p* < 0.05, ** for *p* < 0.01, *** for *p* < 0.001 and **** for *p* < 0.0001.

**Figure 8 cancers-13-04504-f008:**
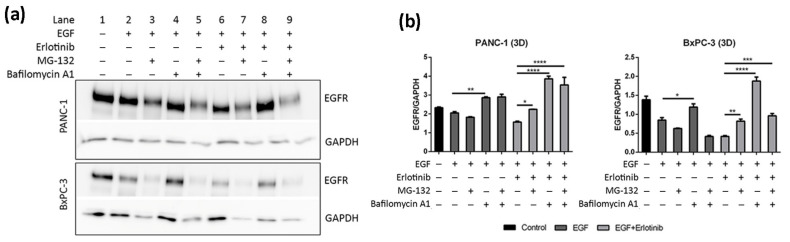
Western blot analysis of the EGFR in PANC-1 and BxPC-3 cells incubated with EGF and erlotinib in the presence of the proteasome (MG-132) and lysosomes (Bafilomycin A1) inhibitors in 0.15% RAD16-I 3D cultures. (**a**) Western blot bands of EGFR; (**b**) densitometry of bands shown in (**a**). GAPDH was used as an internal control. One representative blot is shown. Experiments were repeated three times (N = 3), and statistical differences are indicated as * for *p* < 0.05, ** for *p* < 0.01, *** for *p* < 0.001 and **** for *p* < 0.0001.

**Figure 9 cancers-13-04504-f009:**
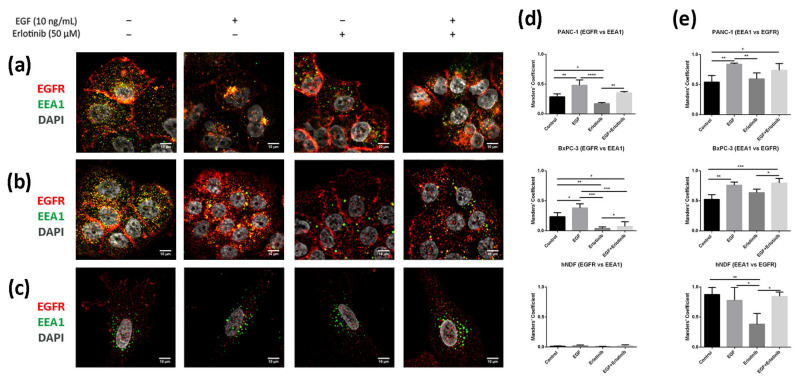
Colocalization analysis of the EGFR and EEA1 (early endosomes) in cells incubated with EGF, erlotinib or both in 2D cultures. EGFR (red) and EEA1 (green) immunofluorescence counterstained with DAPI (grey) in (**a**) PANC-1; (**b**) BxPC-3 and (**c**) hNDF cells. Scale bars represent 10 µm; (**d**) Manders’ colocalization coefficients showing the proportion of EGFR overlapping with EEA1; (**e**) Manders’ colocalization coefficients showing the proportion of EEA1 overlapping with EGFR. Statistical differences are indicated as * for *p* < 0.05, ** for *p* < 0.01, *** for *p* < 0.001 and **** for *p* < 0.0001.

**Figure 10 cancers-13-04504-f010:**
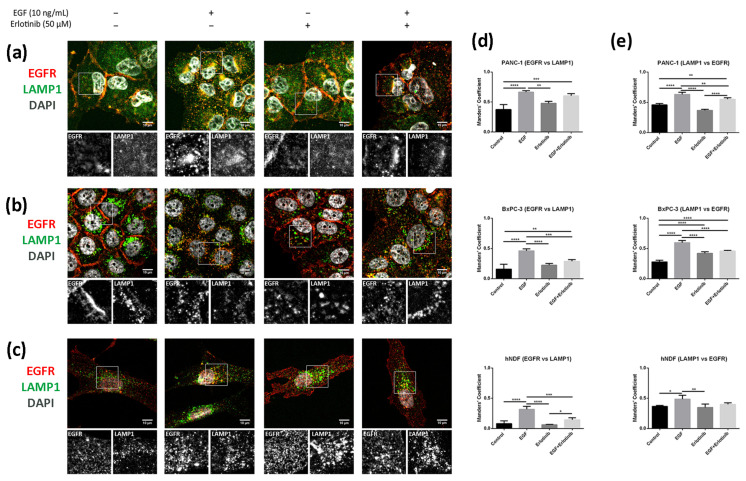
Colocalization analysis of the EGFR and LAMP1 (lysosomes) in cells incubated with EGF, erlotinib or both in 2D cultures. EGFR (red) and LAMP1 (green) immunofluorescence counterstained with DAPI (grey) in (**a**) PANC-1; (**b**) BxPC-3 and (**c**) hNDF cells. Scale bars represent 10 µm. Gray pictures show close-up sections of each marker; (**d**) Manders’ colocalization coefficients showing the proportion of EGFR overlapping with LAMP1; (**e**) Manders’ colocalization coefficients showing the proportion of LAMP1 overlapping with EGFR. Statistical differences are indicated as * for *p* < 0.05, ** for *p* < 0.01, *** for *p* < 0.001 and **** for *p* < 0.0001.

**Figure 11 cancers-13-04504-f011:**
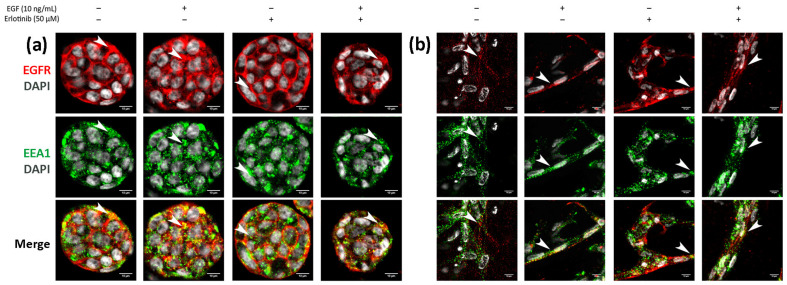
Immunofluorescence of EGFR and EEA1 in (**a**) BxPC-3 cells and (**b**) hNDF incubated with EGF, erlotinib or both in 0.15% RAD16-I 3D cultures. One representative Z plane is shown. Scale bars represent 10 µm. White arrows show EGFR colocalizing with early endosomes.

**Figure 12 cancers-13-04504-f012:**
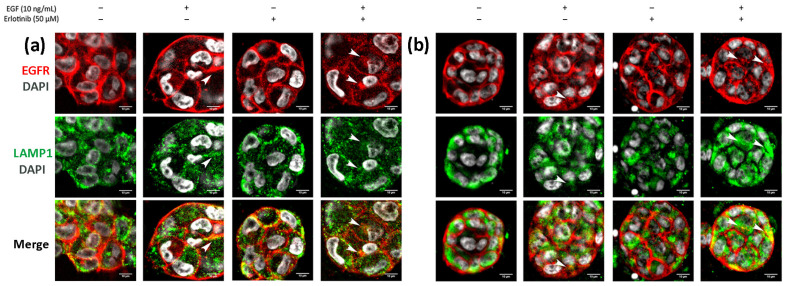
Immunofluorescence of EGFR and LAMP1 in (**a**) PANC-1 and (**b**) BxPC-3 cells incubated with EGF, erlotinib or both in 0.15% RAD16-I 3D cultures. One representative Z plane is shown. Scale bars represent 10 µm. White arrows show EGFR colocalizing with lysosomes.

**Table 1 cancers-13-04504-t001:** Doubling time (h) and erlotinib sensitivity in 2D and 3D cultures.

	Condition	BxPC-3	PANC-1	hNDF
Doubling time (h)	2D	39	30	24.3
	0.15% RAD	ND	63.6	No proliferation
	0.5% RAD	95	62.3	65.3
Erlotinib IC_50_ (µM)	2D	10	45	10
	3D	10	100	10

ND: not determined.

## Data Availability

Data is contained within the article or Appendix A.

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
