# Peer review of "Erlotinib Promotes Ligand-Induced EGFR Degradation in 3D but Not 2D Cultures of Pancreatic Ductal Adenocarcinoma Cells"

_cancers, 2021, doi:10.3390/cancers13184504_

Round 1

Reviewer 1 Report

The paper is an investigation of the 3D culture model of the pancreas for the drug response. The result is clear and understandable. However, the authors should claim this study comparing the related studies. In particular, 3D cancer models combined with biomaterials to mimic the tumor microenvironment have been reported. Related references should be added and described. The paper would be accepted in Cancers when all of the below comments are responded to for revision.

1. Lines 67-85;

3D cancer cell culture is important to mimic the tumor microenvironment. However, there are few references and a little old. Recently, researches on the 3D cell culture to mimic the tumor microenvironment combined with biomaterials as a scaffold has been reported. The authors should introduce the 3D cell culture application quoting the related papers because the 3D cell culture is a keyword in this study.

I suggest some recent papers be quoted for readers’ better understanding.

Review papers

Cancers 202012(10), 2754

Cancers 202012(5), 1305

Research papers

Sci. Rep. 20199, 292

Acta Biomater. 201875, 213–225. 

Tissue Engineering Part C: Methods. 2019, 711-720.

Front. Bioeng. Biotechnol., 11 March 2021 | https://doi.org/10.3389/fbioe.2021.647031

  1. Line 110;

The kinds of cells should be explained, such as human pancreatic cells.

  1. Figure 1;

How about the cell death due to hypoxia, although the cancer cells are more vital in low concentrations of oxygen?

The size is huge, so I am afraid of the problem. The point should be described.

  1.  

Pancreatic cancer is composed of abundant ECM, and the ECM-rich environment affects the poor drug effect. Therefore, the ECM expression level of each 3D model should be investigated.

  1.  

How about the stiffness of each 3D model? The stiffness affects the drug permeability, leading to the difference in drug effects. The authors should explain the points.

Author Response

Open Review

English language and style

( ) Extensive editing of English language and style required
(x) Moderate English changes required
( ) English language and style are fine/minor spell check required
( ) I don't feel qualified to judge about the English language and style

Yes

Can be improved

Must be improved

Not applicable

Does the introduction provide sufficient background and include all relevant references?

( )

( )

(x)

( )

Is the research design appropriate?

(x)

( )

( )

( )

Are the methods adequately described?

(x)

( )

( )

( )

Are the results clearly presented?

(x)

( )

( )

( )

Are the conclusions supported by the results?

(x)

( )

( )

( )

Comments and Suggestions for Authors

The paper is an investigation of the 3D culture model of the pancreas for the drug response. The result is clear and understandable. However, the authors should claim this study comparing the related studies. In particular, 3D cancer models combined with biomaterials to mimic the tumor microenvironment have been reported. Related references should be added and described. The paper would be accepted in Cancers when all of the below comments are responded to for revision.

  1. Lines 67-85; 3D cancer cell culture is important to mimic the tumor microenvironment. However, there are few references and a little old. Recently, researches on the 3D cell culture to mimic the tumor microenvironment combined with biomaterials as a scaffold has been reported. The authors should introduce the 3D cell culture application quoting the related papers because the 3D cell culture is a keyword in this study. I suggest some recent papers be quoted for readers’ better understanding.
  • Review papers: Cancers2020, 12(10), 2754 / Cancers 2020, 12(5), 1305
  • Research papers: Rep.2019, 9, 292 / Acta Biomater. 2018, 75, 213–225. / Tissue Engineering Part C: Methods. 2019, 711-720. / Front. Bioeng. Biotechnol., 11 March 2021 / https://doi.org/10.3389/fbioe.2021.647031

We have added some new references according to the papers suggested by the reviewer (references 19, 20, 21) (lines 90-94)

  1. Line 110; The kinds of cells should be explained, such as human pancreatic cells.

 Thank you for your suggestion. As requested, the origin of these cells has been explained.

  1. Figure 1; How about the cell death due to hypoxia, although the cancer cells are more vital in low concentrations of oxygen? The size is huge, so I am afraid of the problem. The point should be described.

It is a very good point and needs to be clarified. Now, it is true that in the inner part of the 3D construct (the core) oxygen levels are much lower than at the structure periphery. However, results from MTT and Live/Dead assays performed previously in our lab, using different cell lines as well as primary cultures and similar 3D constructs size, demonstrate that oxygen levels at the core of RAD16-I 3D constructs are low but nevertheless do not affect cell viability. The basic principle behind this behavior is that oxygen pression at the incubator is extremely high (around 20%), which is non-physiologic.  Thus, oxygen pressure in 3D cultures, as the reviewer indicated, are much lower and in addition, a gradient (surface to center core) is generated by the equilibrium between diffusion and cellular consumption. All cells actually migrate to the optimum oxygen pressure required according to their particular physiology. Cancer cells in general are in much better shape into these low oxygen pressure but also normal cells, like skin fibroblast, do so. Some examples of cell viability pictures can be found in the following papers:

  • Betriu, N., Jarrosson-Moral, C., & Semino, C. E. (2020). Culture and differentiation of human hair follicle dermal papilla cells in a soft 3D self-assembling peptide scaffold. Biomolecules10(5), 684.
  • Recha‐Sancho, L., & Semino, C. E. (2016). Heparin‐based self‐assembling peptide scaffold reestablish chondrogenic phenotype of expanded de‐differentiated human chondrocytes. Journal of Biomedical Materials Research Part A104(7), 1694-1706.
  • Castells-Sala, C., Recha-Sancho, L., Llucià-Valldeperas, A., Soler-Botija, C., Bayes-Genis, A., & Semino, C. E. (2016). Three-dimensional cultures of human subcutaneous adipose tissue-derived progenitor cells based on RAD16-I self-assembling peptide. Tissue Engineering Part C: Methods22(2), 113-124.
  • Alemany-Ribes, M., García-Díaz, M., Busom, M., Nonell, S., & Semino, C. E. (2013). Toward a 3D cellular model for studying in vitro the outcome of photodynamic treatments: accounting for the effects of tissue complexity. Tissue Engineering Part A19(15-16), 1665-1674.

  1. Pancreatic cancer is composed of abundant ECM, and the ECM-rich environment affects the poor drug effect. Therefore, the ECM expression level of each 3D model should be investigated.

As mentioned by the reviewer, pancreatic cancer is characterized by a dense and desmoplastic stroma, and therefore, 3D cancer cell models that can reproduce this microenvironment are very valuable. Here, the cells are cultured in a 3D configuration within a synthetic and inert matrix, until they produce and decorate their own environment. It is very interesting to us to investigate and characterize this ECM components production by the cells, and this is actually one of the research areas in our lab. However, the aim of this article was not characterizing the ECM production by the cells under each 3D condition (different stiffness), but to study the trafficking and degradation of the EGFR. Now, saying this, it could be also important to see in the future how different stiffness as well as ECM composition may or may not affect EGFR trafficking.

  1. How about the stiffness of each 3D model? The stiffness affects the drug permeability, leading to the difference in drug effects. The authors should explain the points.

In our case we performed MTT assay to determine cell viability after erlotinib incubation under both stiffness conditions including 0.15% and 0.5% RAD16-I (which represent about 120 Pa and 1500 Pa, respectively) with PDAC cell lines. Interestingly, we found no differences, suggesting that under our experimental conditions, drug permeability is not affected by stiffness (or at least not in the range of stiffness used in this paper).

Reviewer 2 Report

This is a very interesting paper investigating the impact of (i) type of culture (2D vs 3D)  (ii) type of cells (two different pancreatic cancer cell lines vs human dermal fibroblasts (iii) stiffness levels in 3D on the trafficking/endocytosis/degradation of EGFR resulting from TKI treatment.

I find the paper of high standard, very well written, the images very clear and the results sound and interesting.

I suggest publication of this manuscript which is highly relevant both for the special issue and the journal (Cancers) in general.

I have some minor points for the authors to consider:

-Simple summary: The authors might want to review the summary as at times it is not as simple/layman. For example the authors might want to clearly explain what ‘stresses’ are (they describe it elsewhere in the scientific text but an example or two would be useful here) or the authors might want to elaborate on the term ‘dimensionality’ in 3D cultures in a way that a non-expert would understand.

-Abstract: There is a small typographical error: ‘this is why are rising’ should be replaced with ‘this is why they are rising’.

-Introduction: The authors cite some 3D papers on cancers other than PDAC (e.g. glioblastoma) which is absolutely fine, however, considering that the focus on the work is on PDAC they might want to expand a bit their reflection of the state of the art with some PDAC focused papers. I appreciate that this might not be obvious for 3D PDAC studies of EGFR trafficking due to the novelty of the work, but maybe the authors could consider such expansion in the part where they discuss the benefits of 3D models including scaffolds.

Some recent examples of PDAC 3D work in either spheroids or scaffolds involve the following:

Spheroids: https://pubmed.ncbi.nlm.nih.gov/23446043/

Scaffolds: https://pubmed.ncbi.nlm.nih.gov/32391339/

                  https://pubmed.ncbi.nlm.nih.gov/25482337/

-Dermal fibroblast choice as control: I appreciate the author’s choice of control, especially in relation and follow up to their previous work. However, the authors refer to this control as a ‘comparison to a healthy compartment’. If we refer to the tumour microenvironment itself wouldn’t it be reasonable to consider activated stellate cells as a control of the stromal compartment for PDAC? Is this something the authors would consider in the future? Maybe the authors can re-phrase this part suggesting that the dermal fibroblast is indeed a good surrogate of a ‘healthy cell control’ but they might want to avoid linking the dermal fibroblasts to the PDAC stroma specifically.

-Conclusions: The conclusions are sound and focused on the EGFR trafficking mechanisms. Would the authors expect differences in trafficking trends following TKI treatment in 3D systems of even higher complexity, e.g. scaffold based or for different ECM configurations? The authors might want to add a short comment on that as this would be useful for the tissue engineering community.

Author Response

Open Review

English language and style

( ) Extensive editing of English language and style required
( ) Moderate English changes required
(x) English language and style are fine/minor spell check required
( ) I don't feel qualified to judge about the English language and style

Yes

Can be improved

Must be improved

Not applicable

Does the introduction provide sufficient background and include all relevant references?

( )

(x)

( )

( )

Is the research design appropriate?

(x)

( )

( )

( )

Are the methods adequately described?

(x)

( )

( )

( )

Are the results clearly presented?

(x)

( )

( )

( )

Are the conclusions supported by the results?

( )

(x)

( )

( )

Comments and Suggestions for Authors

This is a very interesting paper investigating the impact of (i) type of culture (2D vs 3D)  (ii) type of cells (two different pancreatic cancer cell lines vs human dermal fibroblasts (iii) stiffness levels in 3D on the trafficking/endocytosis/degradation of EGFR resulting from TKI treatment.

 I find the paper of high standard, very well written, the images very clear and the results sound and interesting.

I suggest publication of this manuscript which is highly relevant both for the special issue and the journal (Cancers) in general.

I have some minor points for the authors to consider:

-Simple summary: The authors might want to review the summary as at times it is not as simple/layman. For example the authors might want to clearly explain what ‘stresses’ are (they describe it elsewhere in the scientific text but an example or two would be useful here) or the authors might want to elaborate on the term ‘dimensionality’ in 3D cultures in a way that a non-expert would understand.

 As suggested by the reviewer, we have added some examples of stresses that induce EGFR endocytosis in the simple summary. In addition, we clarified our definition of 3D scaffolds, as truly three-dimensional environment due to their nanostructured properties, where cells can growth, proliferate, migrate and extent processes in their 3D space (see line 14)

-Abstract: There is a small typographical error: ‘this is why are rising’ should be replaced with ‘this is why they are rising’.

 Thank you very much. This error has been revised.

-Introduction: The authors cite some 3D papers on cancers other than PDAC (e.g. glioblastoma) which is absolutely fine, however, considering that the focus on the work is on PDAC they might want to expand a bit their reflection of the state of the art with some PDAC focused papers. I appreciate that this might not be obvious for 3D PDAC studies of EGFR trafficking due to the novelty of the work, but maybe the authors could consider such expansion in the part where they discuss the benefits of 3D models including scaffolds. Some recent examples of PDAC 3D work in either spheroids or scaffolds involve the following:

Spheroids:https://pubmed.ncbi.nlm.nih.gov/23446043/

Scaffolds:https://pubmed.ncbi.nlm.nih.gov/32391339/,  https://pubmed.ncbi.nlm.nih.gov/25482337/

As mentioned by the reviewer, we could not find previous work on EGFR trafficking in PDAC cells, neither in 2D nor in 3D cultures.  However, we appreciate this suggestion and we have added some new information about 3D culture of PDAC cells (lines 93-110, refs 20-23, 26, 29)       

-Dermal fibroblast choice as control: I appreciate the author’s choice of control, especially in relation and follow up to their previous work. However, the authors refer to this control as a ‘comparison to a healthy compartment’. If we refer to the tumour microenvironment itself wouldn’t it be reasonable to consider activated stellate cells as a control of the stromal compartment for PDAC? Is this something the authors would consider in the future? Maybe the authors can re-phrase this part suggesting that the dermal fibroblast is indeed a good surrogate of a ‘healthy cell control’ but they might want to avoid linking the dermal fibroblasts to the PDAC stroma specifically.

Thank you for your suggestion. In this work, we use dermal fibroblasts (hNDF) and not pancreatic fibroblasts or pancreatic stellate cells because of the availability of the first ones and because hNDF are easier to expand and culture and they don’t need special medium supplements. Indeed, this fibroblasts control was meant to be a control for healthy cells in general, and as suggested by the reviewer, we have clarified this in the text (line 239).

-Conclusions: The conclusions are sound and focused on the EGFR trafficking mechanisms. Would the authors expect differences in trafficking trends following TKI treatment in 3D systems of even higher complexity, e.g. scaffold based or for different ECM configurations? The authors might want to add a short comment on that as this would be useful for the tissue engineering community.

Thank you for your suggestion. It is actually a very important issue and yes, we have added a short comment as suggested. In fact, PDAC ECM is complex and future research might focus on the effect that different components, such as collagens and hyaluronic acid, could have on EGFR trafficking.

Reviewer 3 Report

The authors shall address the following comments:

  1. In the introduction, lanes 78-85 references on biomaterial scaffold and 3D cancer cell models should be added: https://doi.org/10.1098/rsif.2016.0877; https://doi.org/10.1002/adhm.201700980; https://doi.org/10.3390/pharmaceutics13070963.
  2. Legend to Fig 1: please specify “MTT assay at two different time points (day1, day 7)”
  3. Page 7/278 and page 8/284 I can find Fig. 3 e
  4. To make the results easier to read, I would combine, if possible, the results of 2D cell cultures in a Table and move the Figures 3-4-5 to the supp mat.
  5. Page 9/331-336 there is no comment on the treatment of 3D cultures with Erlotinib alone
  6. Legend to fig.7 (c) and (d) “shown” instead of “showed”
  7. Fig 7b hNDF western lane 8

Author Response

Open Review

English language and style

( ) Extensive editing of English language and style required
( ) Moderate English changes required
(x) English language and style are fine/minor spell check required
( ) I don't feel qualified to judge about the English language and style

Yes

Can be improved

Must be improved

Not applicable

Does the introduction provide sufficient background and include all relevant references?

( )

(x)

( )

( )

Is the research design appropriate?

(x)

( )

( )

( )

Are the methods adequately described?

(x)

( )

( )

( )

Are the results clearly presented?

( )

( )

(x)

( )

Are the conclusions supported by the results?

(x)

( )

( )

( )

Comments and Suggestions for Authors

The authors shall address the following comments:

  1. In the introduction, lanes 78-85 references on biomaterial scaffold and 3D cancer cell models should be added: https://doi.org/10.1098/rsif.2016.0877; https://doi.org/10.1002/adhm.201700980; https://doi.org/10.3390/pharmaceutics13070963.

References suggested by the reviewer on biomaterial scaffold and 3D cancer models were added (Refs 16, 17)

  1. Legend to Fig 1: please specify “MTT assay at two different time points (day1, day 7)”

            As requested, we have changed the legend.

  1. Page 7/278 and page 8/284 I can find Fig. 3 e

Thank you, this mistake has been corrected to Fig 3c

  1. To make the results easier to read, I would combine, if possible, the results of 2D cell cultures in a Table and move the Figures 3-4-5 to the supp mat.

We are aware that in some points pictures can be a little bit repetitive, since for example the results between both cancer cell lines are very similar. However, we consider that these pictures are important for the results to be easily understandable, and we think that moving it into the supplementary materials section could confuse the reader.

  1. Page 9/331-336 there is no comment on the treatment of 3D cultures with Erlotinib alone

We have added the following: “ Under control conditions and in the presence of erlotinib alone, EGFR staining was peripheral and well-defined…”

  1. Legend to fig.7 (c) and (d) “shown” instead of “showed”

This typographic mistake has been changed.

  1. Fig 7b hNDF western lane 8

We do not see any issue with this blot. It is true that gel loading presents some differences, by looking at GAPDH levels, but in any case, comparative quantification was plotted at the side. In other words, protein levels can be relatively compared between normalized samples.

Reviewer 4 Report

In this paper by Betriu et al. titled “Erlotinib promotes ligand-induced EGFR degradation in 3D but not 2D cultures of pancreatic ductal adenocarcinoma cells”, the authors use RAD16-I, a synthetic self-assembling peptide, as a scaffold to culture pancreatic ductal adenocarcinoma (PDAC) cells (BxPC3 and PANC-1) or normal human fibroblasts (NHFs) in 3D. The authors compare the effect of the epidermal growth factor receptor (EGFR) inhibitor erlotinib on EGFR internalization and degradation in 3D vs normal 2D culture. The authors report differences between 3D vs 2D culture, as well as differences between PDAC cells vs NHFs. I overall felt that the study was very interesting and touches upon a very important topic in basic cell biology as well as cancer biology. However, I was also of the impression that the manuscript, as is, falls short of demonstrating the central claims being made. I detail my concerns below.

Major comments:

1, Table 1: It would be better to show the whole dose-response curve rather than simply report IC50s. Also, I think details regarding IC50 calculation are missing from the materials & methods section.

2, Page 5, lines 236-237: Please specify which data the authors are basing this claim upon. Also, “cell phenotype” is a very broad term, so the authors should specific which cell phenotype they have assessed and are referring to.

3, Figure 1, macroscopic view: Is there data for BxPC-3? Also, quantification would help visualize inter-sample variability.

4, Figure 2 (a) and (b): Providing images at same magnification for 2D (a) and 3D (b) would ease comparison between the two culture conditions. Specifically, higher magnification images for the 3D condition would help in better assessing localization as the images provided are somewhat too small to look at intracellular localization.

5, Figure 2 (c) and (d): In the EGFR blot (c) for PANC-1, I see a clear decrease in EGFR levels between 0.15% and 0.5% conditions, but the quantification data (d) says otherwise. Please explain.

6, Figure 5: Same analysis should be performed as PANC-1 (Figure 3) and BxPC-3 (Figure 4) for a fair comparison between the different cell types.

7, Figure 6, 11, and 12: The data should be quantified to show differences in localization. For Figure 6, to ease comparison with the experiments performed in 2D, the experiment should be performed in similar fashion (co-stain with integrin β1). Quantification is especially important since localization is essentially a quantitative feature (localization changes are usually not all-or-nothing, but more gradual).

8, Western blots: Overall, I felt that the band intensities of the loading control were inconstant in numerous blots presented in the manuscript. While I do not insist upon perfectly constant loading controls (I understand that such an expectation is un-realistic), it does become problematic when it is inconstant to such an extent that changes in the denominator (band intensity of GAPDH) affects trends in the numerator (band intensity of EGFR) as one can no longer rule out the possibility that it is primarily changes in GAPDH expression that is driving the overall change in GAPDH normalized protein levels (EGFR/GAPDH). A case in point is Figure 7. The authors claim that “the presence of erlotinib in EGF-incubated fibroblasts in 3D, not only did not induce the degradation of the receptor but prevented it” (page 11, lines 366-367). When one looks at the band intensities for EGFR and GAPDH (Figure 7(b) lanes 5-8), one sees that the band intensity of EGFR is most faint in lane 8. However, since the GAPDH band is also most faint in lane 8, these changes cancel out. I am not convinced by this data that the EGF + erlotinib combination in NHFs prevents degradation. A similar problem is seen in Figure 8.

Minor comments:

1, page 2, lines 78-80: A reference to accompany this claim would help readers who are not entirely familiar with 3D cell culture.

2, page 2, lines 80-83, lines 88-90: References to accompany these claims are requested.

3, page 3, line 130: P/S is an undefined abbreviation.

4, page 3, line 136: Please specify what the cells were seed on for 2D assays. Tissue-culture plastic? Glass (if so, coated with what)?

5, page 4, line 156, line 157: A should be Alexa Fluor.

6, page 5, lines 197-198: The authors mention conditions been tested in triplicate. However, in section 2.6 (page 4, Line 178), the authors mention >5 images. Does this mean >5 images for triplicates per experiment, performed three times independently? How was the data then aggregated to give the final graphs? Please provide details.

7, Figures 3-5, Figures 9 and 10, and Figure 11 and 12 might as well be one figure each.

8, page 11, line 392: The authors mention “synergic” treatment, but “combinatorial” would be a better term as there is no experimental evidence provided to demonstrate synergy (rather than a simple additive effect).

9, Overall, the scale bars provided seemed too small throughout most of the manuscript.

Author Response

Open Review

English language and style

( ) Extensive editing of English language and style required
(x) Moderate English changes required
( ) English language and style are fine/minor spell check required
( ) I don't feel qualified to judge about the English language and style

Yes

Can be improved

Must be improved

Not applicable

Does the introduction provide sufficient background and include all relevant references?

( )

(x)

( )

( )

Is the research design appropriate?

( )

( )

(x)

( )

Are the methods adequately described?

( )

(x)

( )

( )

Are the results clearly presented?

( )

( )

(x)

( )

Are the conclusions supported by the results?

( )

( )

(x)

( )

Comments and Suggestions for Authors

In this paper by Betriu et al. titled “Erlotinib promotes ligand-induced EGFR degradation in 3D but not 2D cultures of pancreatic ductal adenocarcinoma cells”, the authors use RAD16-I, a synthetic self-assembling peptide, as a scaffold to culture pancreatic ductal adenocarcinoma (PDAC) cells (BxPC3 and PANC-1) or normal human fibroblasts (NHFs) in 3D. The authors compare the effect of the epidermal growth factor receptor (EGFR) inhibitor erlotinib on EGFR internalization and degradation in 3D vs normal 2D culture. The authors report differences between 3D vs 2D culture, as well as differences between PDAC cells vs NHFs. I overall felt that the study was very interesting and touches upon a very important topic in basic cell biology as well as cancer biology. However, I was also of the impression that the manuscript, as is, falls short of demonstrating the central claims being made. I detail my concerns below.

Major comments:

1, Table 1: It would be better to show the whole dose-response curve rather than simply report IC50s. Also, I think details regarding IC50 calculation are missing from the materials & methods section.

The whole dose-response curve for erlotinib in 2D and 3D cultures were included as a supplementary figure (Figure S1). Also, methods on IC50 calculation were added in section 2.4.

2, Page 5, lines 236-237: Please specify which data the authors are basing this claim upon. Also, “cell phenotype” is a very broad term, so the authors should specific which cell phenotype they have assessed and are referring to.

By visual inspection of the photographical data (Phalloidin staining – actin cytoskeleton) we arrived to the conclusion that each cell type presents a distinct general phenotype (shape). As stated clearly in the manuscript: “Regarding cell phenotype, the epithelial cell line BxPC-3 grew forming round or oval clusters (Figure 1b, left) while PANC-1 cells formed more grape-like spheres (Figure 1b, middle). We did not find a correlation between matrix stiffness and cell phenotype or colony size in the tumor cells analyzed”.

3, Figure 1, macroscopic view: Is there data for BxPC-3? Also, quantification would help visualize inter-sample variability.

BxPC-3 shown the same behavior as PANC-1 cells (no contraction of the matrix and therefore no size change of the construct along culture time). By looking at the pictures presented in the manuscript it can be qualitatively seen that cells from epithelial origin that grow forming clusters in 3D (such as PDAC cells and tumor cells in general) do not contract the gels in which they are cultured. Macroscopic changes of the 3D construct only happen with fibroblasts and other cells from mesenchymal origin, because they can form a network and thus contract the gel. Some examples of macroscopic changes in RAD16-I 3D constructs seeded with mesenchymal cells from different origins are provided:

  • Betriu, N., Jarrosson-Moral, C., & Semino, C. E. (2020). Culture and differentiation of human hair follicle dermal papilla cells in a soft 3D self-assembling peptide scaffold. Biomolecules10(5), 684.
  • Bussmann, B. M., Reiche, S., Marí‐Buyé, N., Castells‐Sala, C., Meisel, H. J., & Semino, C. E. (2016). Chondrogenic potential of human dermal fibroblasts in a contractile, soft, self‐assembling, peptide hydrogel. Journal of tissue engineering and regenerative medicine10(2), E54-E62.
  • Castells-Sala, C., Recha-Sancho, L., Llucià-Valldeperas, A., Soler-Botija, C., Bayes-Genis, A., & Semino, C. E. (2016). Three-dimensional cultures of human subcutaneous adipose tissue-derived progenitor cells based on RAD16-I self-assembling peptide. Tissue Engineering Part C: Methods22(2), 113-124.
  • Marí-Buyé, N., Luque, T., Navajas, D., & Semino, C. E. (2013). Development of a three-dimensional bone-like construct in a soft self-assembling peptide matrix. Tissue Engineering Part A19(7-8), 870-881.

4, Figure 2 (a) and (b): Providing images at same magnification for 2D (a) and 3D (b) would ease comparison between the two culture conditions. Specifically, higher magnification images for the 3D condition would help in better assessing localization as the images provided are somewhat too small to look at intracellular localization.

As suggested by the reviewer, we changed Figure 2b with higher magnification images (3D cultures)

5, Figure 2 (c) and (d): In the EGFR blot (c) for PANC-1, I see a clear decrease in EGFR levels between 0.15% and 0.5% conditions, but the quantification data (d) says otherwise. Please explain.

Looking at the EFGR band densities only can be misleading. Now, quantification data presented in Figure 2d has been obtained from three different blots (N=3). For each sample, the normalization of EGFR/GAPDH has been made. Then, normalized values obtained from soft and stiff were used to compare differences between them. Therefore, the results are presented as an average of the ratio between soft and stiff cultures from three independent experiments.

6, Figure 5: Same analysis should be performed as PANC-1 (Figure 3) and BxPC-3 (Figure 4) for a fair comparison between the different cell types.

PANC-1 and BxPC-3 cells are pancreatic cancer cell lines from epithelial tissue, while hNDF cells are mesenchymal cells with a cytoskeleton strongly formed of stress fibers. The phenotype (epithelial vs mesenchymal) of these types of cells really diverges from each other from the point of view of the general morphology as well as their biomolecular components composition and cellular distribution. Fibroblasts present a spread morphology, with slight β1-integrin distribution at the cell periphery of the cell and mainly associated stress fibers (forming focal adhesions along the stress fibers), which is very different from PDAC cells. Moreover, in PDAC cells β1-integrin staining is strong and clearly peripheric, as expected in epithelial cell types. Therefore, in this case, fibroblasts and PDAC cells are not comparable in terms of co-localization of EGFR and β1-integrin.

7, Figure 6, 11, and 12: The data should be quantified to show differences in localization. For Figure 6, to ease comparison with the experiments performed in 2D, the experiment should be performed in similar fashion (co-stain with integrin β1). Quantification is especially important since localization is essentially a quantitative feature (localization changes are usually not all-or-nothing, but more gradual).

In figure 6, β1-integrin staining is not shown because it impeded the clearly visualization of EGFR staining. On the other hand, co-localization quantification in 3D has not been possible to perform due to the inherent background associated to immunofluorescence technique in scaffold-based 3D cultures. In our particular case, we are using very advance confocal microscopy techniques (Wide-field, Thunder) and as far as we understand, colocalization in 3D structures is very difficult task and quantification is even more difficult and impossible for certain samples. Indeed, 3D cultures are a more realistic approach than 2D cultures, but as such, also present some drawbacks, being one of them that they are more difficult to analyze and therefore quantify. In addition, it is well known that 3D systems based on spheroids are much easier to visualize by 3D colocalization techniques (and quantify) than 3D systems developed in matrices. Moreover, 3D systems developed in synthetic self-assembling peptides have many advantages compared to natural-based scaffolds (like collagens) but some disadvantages, which is the fact that some antibodies or combination of them might generate background. Finally, while we agree with the reviewer about this issue it is also true that quantitative colocalization of EGFR and β1-integrin do not affect the main results observed and, therefore, the main conclusions. The fact that the EGFR cellular trafficking involves its degradation is quantified by western blots and microscopic localization is a control, confirming its colocalization with different subcellular markers.

8, Western blots: Overall, I felt that the band intensities of the loading control were inconstant in numerous blots presented in the manuscript. While I do not insist upon perfectly constant loading controls (I understand that such an expectation is un-realistic), it does become problematic when it is inconstant to such an extent that changes in the denominator (band intensity of GAPDH) affects trends in the numerator (band intensity of EGFR) as one can no longer rule out the possibility that it is primarily changes in GAPDH expression that is driving the overall change in GAPDH normalized protein levels (EGFR/GAPDH). A case in point is Figure 7. The authors claim that “the presence of erlotinib in EGF-incubated fibroblasts in 3D, not only did not induce the degradation of the receptor but prevented it” (page 11, lines 366-367). When one looks at the band intensities for EGFR and GAPDH (Figure 7(b) lanes 5-8), one sees that the band intensity of EGFR is most faint in lane 8. However, since the GAPDH band is also most faint in lane 8, these changes cancel out. I am not convinced by this data that the EGF + erlotinib combination in NHFs prevents degradation. A similar problem is seen in Figure 8.

When performing protein extraction, and in contrast to 2D cultures, cells in 3D scaffolds are lysed without detaching them from the surrounding ECM prior to adding cell lysis buffer. Moreover, FBS proteins from the 3D construct cannot be completely removed by washing prior to lysis. This contamination with non-cellular proteins also interferes with BCA assay to measure protein concentration (1). Moreover, it has been reported the presence of deposition of ECM proteins (such as fibronectin, collagen I and laminin) by fibroblasts and cancer cells in synthetic matrices (2). Therefore, protein concentration, from cellular origin, in 3D cell lysates may be lower than in 2D lysates due to the dilution with remaining FBS proteins and ECM proteins deposited by cells, which can introduce variability between samples (specifically when looking GAPDH bands). Moreover, it has been described that substrate stiffness influences and increases the deposition of some ECM proteins such as laminin (3), which could also dilute the total cellular protein content. Moreover, FBS proteins in stiffer gels may be harder to elute from the 3D construct due to lower diffusion. In particular, with fibroblasts and additional effect also interferes with protein construct extraction which is the fact that constructs in soft matrices contracts, becoming stiffer, and therefore increasing the difficulty of protein extraction. All these facts could explain the weaker GAPDH band obtained for fibroblasts in stiff 3D cultures. Now, by looking carefully Figure 7, all these effects can be easily observed and correlates well with the protein bands detected.

On the other hand, in Figure 8 two cancer cell lines are compared without the presence of fibroblasts. Interestingly, differences between GAPDH bands are observed with the BxPC-3 cell line specifically when cells were incubated in presence of MG-132 (a general proteasome inhibitor). We think that incubation with MG-132 in BxPC-3 cells could affect GAPDH expression, as well as other proteins in general, since this inhibitor might have a general effect on protein synthesis-degradation status which, for some reason, is more evident in BxPC-3 than in PANC-1 cells. We are confident with the results since all experiments were repeated, and the same band pattern was obtained.

  • Eke, I., Hehlgans, S., Zong, Y., & Cordes, N. (2015). Comprehensive analysis of signal transduction in three-dimensional ECM-based tumor cell cultures. Journal of biological methods2(4), e31.
  • Malakpour-Permlid, A., Buzzi, I., Hegardt, C., Johansson, F., & Oredsson, S. (2021). Identification of extracellular matrix proteins secreted by human dermal fibroblasts cultured in 3D electrospun scaffolds. Scientific reports11(1), 1-18.
  • Eisenberg, J. L., Safi, A., Wei, X., Espinosa, H. D., Budinger, G. S., Takawira, D., Hopkinson, S. B., & Jones, J. C. (2011). Substrate stiffness regulates extracellular matrix deposition by alveolar epithelial cells. Research and reports in biology2011(2), 1–12.

Minor comments:

1, page 2, lines 78-80: A reference to accompany this claim would help readers who are not entirely familiar with 3D cell culture.

New references have been added

2, page 2, lines 80-83, lines 88-90: References to accompany these claims are requested.

New references have been added

3, page 3, line 130: P/S is an undefined abbreviation.

This abbreviation has been defined

4, page 3, line 136: Please specify what the cells were seed on for 2D assays. Tissue-culture plastic? Glass (if so, coated with what)?

Cells were seeded in uncoated tissue culture-treated plastic 48 well plates

5, page 4, line 156, line 157: A should be Alexa Fluor.

Changed to Alexa Fluor as requested

6, page 5, lines 197-198: The authors mention conditions been tested in triplicate. However, in section 2.6 (page 4, Line 178), the authors mention >5 images. Does this mean >5 images for triplicates per experiment, performed three times independently? How was the data then aggregated to give the final graphs? Please provide details.

Co-localization coefficients were obtained after analyzing at least 5 random images from three independent experiments (see section 2.6 Image Analysis).

7, Figures 3-5, Figures 9 and 10, and Figure 11 and 12 might as well be one figure each.

We decided to present these results as separated pictures because it permitted bigger images and a clearer visualization of immunofluorescence pictures. Also, if these pictures were merged together, the legend would be in a separate page and we consider that this is uncomfortable for the reader.

8, page 11, line 392: The authors mention “synergic” treatment, but “combinatorial” would be a better term as there is no experimental evidence provided to demonstrate synergy (rather than a simple additive effect).

Ok, changed to combinatorial instead of synergic

9, Overall, the scale bars provided seemed too small throughout most of the manuscript.

We have added in each figure legend “Scale bars represent X µm”

Round 2

Reviewer 4 Report

Thank you for the revision. Some changes have been noted, and the manuscript is improved. However, some concerns remain.

Though I appreciate the authors' explanations of why protein extraction from the 3D cell culture system is difficult (and I understand these points as I, too, work with 3D cell culture systems as well), it does not address the concerns I raised in my previous report. As the authors mention in their response regarding MG132, specific treatments (or more generally, culture conditions) can alter GAPDH expression as well. As such, as I also pointed out in my previous report, changes in the value of protein of interest/GAPDH can be ascribed to both 1) changes in expression of the protein of interest, as well as 2) changes in GAPDH expression.

Author Response

Reviewer 4 (Round 2)

Thank you for the revision. Some changes have been noted, and the manuscript is improved. However, some concerns remain.

Though I appreciate the authors' explanations of why protein extraction from the 3D cell culture system is difficult (and I understand these points as I, too, work with 3D cell culture systems as well), it does not address the concerns I raised in my previous report. As the authors mention in their response regarding MG132, specific treatments (or more generally, culture conditions) can alter GAPDH expression as well. As such, as I also pointed out in my previous report, changes in the value of protein of interest/GAPDH can be ascribed to both 1) changes in expression of the protein of interest, as well as 2) changes in GAPDH expression.

Response:

In agreement with the reviewer comment, in our particular case, we detected cellular response in presence of MG-132 changes in the amount of the protein of interest (EGFR) as well as changes in the expression of GAPDH in BxPC-3 cell line, mainly. We assume that MG-132 is generally affecting protein levels (at least EGFR and GAPDH) and this is why the ratio EGFR/GAPDH can be used to compare the values of this ratio with other tested conditions, in a normalized way. Most importantly, the MG-132 effect does not affect the main conclusions of our work, which indicates that EGFR degradation is in 3D systems through lysosomes, as previously described in 2D cultures.